# SALT : Sharing Attention between Linear layer and Transformer for tabular dataset

## Abstract

Handling tabular data with deep learning models is a challenging problem despite their remarkable success in vision and language processing applications. Therefore, many practitioners still rely on classical models such as gradient boosting decision trees (GBDTs) rather than deep networks due to their superior performance with tabular data. In this paper, we propose a novel hybrid deep network architecture for tabular data, dubbed SALT (Sharing Attention between Linear layer and Transformer). The proposed SALT consists of two blocks: Transformers and linear layer blocks that take advantage of shared attention matrices. The shared attention matrices enable transformers and linear layers to closely cooperate with each other, and it leads to improved performance and robustness. Our algorithm outperforms tree-based ensemble models and previous deep learning methods in multiple benchmark datasets. We further demonstrate the robustness of the proposed SALT with semi-supervised learning and pre-training with small dataset scenarios.

## 1 Introduction

In the fields of vision and natural language processing, deep networks such as CNN, RNN, LSTM and Transformer have gained great popularity with its impressive performance. In particular, Transformer(Vaswani et al., 2017) designed as language model, improves the performance of lots of deep learning models in various domains, so that there are many powerful models based on Transformer (e.g. Devlin et al. (2019), Brown et al. (2020) and Dosovitskiy et al. (2021)).

Although the deep networks are powerful and used in natural language processing and vision, they are sub-optimal for other types of real-world problems that require to use tabular data, such as fraud detection(Luo et al., 2019), product recommendation(Guo et al., 2017) and disease prediction(Koppu et al., 2020). Tabular data has different characteristics from other data. Unlike text data that include vocabularies and words, and image data that include RGB values, the tabular data are usually mixed with different types of complex variables. For example, the tabular data contain continuous variables such as such as age, height,and weight have different ranges of values, and categorical variables which are independent from one another like gender and nationality.

For these reasons, deep learning models were not quite successful with tabular data despite its strength in natural language processing and vision fields. Instead of deep learning models, classical models such as tree-based ensemble models are mainly used for tabular data. However, these classical approaches have some limitations. Continuous learning with real-time data is quite difficult using these classical methods. When the tabular data is high-dimensional sparse, the performance of tree-based methods is degraded. Also, the tree-based ensemble methods do not perform well for multi-modality learning and end-to-end systems.

Therefore, it is an interesting to study a deep learning models for tabular data. Many studies have been attempted not only to overcome the shortcomings of the classical model with a deep learning model but also to overcome its performance. (Arik & Pfister, 2020), (Huang et al., 2020) and (Somepalli et al., 2021).

In this paper, we propose a new hybrid deep learning model architecture, named as SALT (Sharing Attention between Linear layer and Transformer). We summarize the contributions of our paper as follows :

- SALT shares the attention matrices between two blocks. Sharing the attention matrix allows to learn two blocks strongly and effectively. We demonstrate that sharing attention matrices improves the performance better.
- SALT introduce the improved embedding method for continuous variables. We demonstrate that this method performs better than others.
- SALT has four types of variants depending on which block is used and in what direction the attention matrix is shared. Each variant shows strength in different data.
- SALT outperforms the other models on average over a variety of benchmark datasets. It also performs well even in small data environments via semi-supervised and self-supervised learning

## 2 RELATED WORK

### 2.1 TREE-BASED MODELS

Decision trees (Quinlan, 1986) are well-known for their high predictive performance compared to computational complexity. In addition, it has the strength of having explanatory power in units of variables with most statistical information gain. However, decision trees are likely to work well only on specific data because their decision boundaries are perpendicular to the data axis. To improve this drawback and performance, there are many ensemble models of decision trees such as Random forest (Breiman, 2001) and GBDTs(Gradient Boosting Decision Trees). Especially, GBDTs methods such as XGBoost(Chen & Guestrin, 2016), LightGBM(Ke et al., 2017) and CatBoost(Dorogush et al., 2018) are commonly used with powerful performance in lots of machine learning competitions and industrial sites.

### 2.2 DEEP LEARNING MODELS FOR TABULAR DATA

There are several studies on deep networks to overcome the limitations of tree-based models and outperform the performance of GBDTs. Especially there are deep learning models based on attention mechanism. TabNet(Arik & Pfister, 2020), for example, is designed to learn similarly to a decision trees, and it has interpretability with the attentive layer. It shows better performance than GBDTs in some dataset. TabTransformer(Huang et al., 2020) is designed based on Transformer and has contextual embedding values. However, it embeds only categorical variables, so that there is a limitation for continuous variables. There are some models that improve the limitation of TabTransformer(Song et al., 2018),(Somepalli et al., 2021). They have the contextual embedding values of not only categorical variables but also continuous variables. Especially a model called SAINT(Somepalli et al., 2021) introduces inter-sample self-attention method and shows powerful performance. But it has a fatal disadvantage that training costs are high.

### 2.3 TRANSFORMER

Self-attention is the core module of Transformer(Vaswani et al., 2017). The self-attention consists of three parameter matrices : $K$(keys), $Q$(queries), and $V$(values). Formally, input embedding values $X \in \mathbb{R}^{n \times d}$ of $n$ features of dimensions $d$, are projected using $W_Q \in \mathbb{R}^{d \times d_q}$, $W_Q \in \mathbb{R}^{d \times d_q}$, and $W_Q \in \mathbb{R}^{d \times d_q}$ to extract feature representations $Q$, $K$, and $V$. With $Q$, $K$, and $V$ , self-attention can be written as,

$$Q = XW_Q, K = XW_K, V = XW_V \tag{1}$$

$$\text{Attention}(Q, K, V) = \text{Softmax}(\frac{QK^T}{\sqrt{d_k}})V \tag{2}$$

MHA(Multi Head Attention) is having multiple attention heads. Multi head allows the attention matrix to have the abundant representations(Michel et al., 2019),(Voita et al., 2019). Each head attention has different $Q$, $K$, and $V$ weight matrices and calculates the attention values with the

equation 2. All heads are then concatenated and multiplied by the weight matrix to generate the final output of the layer.

$$\text{MHA}(Q, K, V) = \text{Concat}(head_1, ..., head_h)W^O \tag{3}$$

where $head_i = \text{Attention}(Q, K, V)$ and $W^O$ is weighted matrix for final output. The dimension $d_h$ of each head is typically given as $d/N_h$.

### 2.4 GATING MECHANISM

The gating mechanism is an effective method of learning by controlling the information flow path of the LSTM (Hochreiter & Schmidhuber, 1997). However, the more layers of these mechanisms are stacked, the more likely the gradient is vanishing. As an improved mechanism, gate linear units(GLU) is introduced by (Dauphin et al., 2017). GLU has been used in many deep learning models and shows the better performance (Arik & Pfister, 2020),(Shazeer, 2020). To briefly explain the GLUs, they divide the input in half, take an activation function on one side, and multiply by element with the other. Therefore, the output dimension is half the input dimension.

$$h(x) = (W_1 X + b_1) \otimes \sigma(W_2 X + b_2) \tag{4}$$

The gating mechanism is used to control the information flow slightly similar to the self-attention mechanism. The main difference between gating mechanism and self-attention is that gating mechanism controls only the bandwidth of a single element, while self-attention considers information of two different elements.

## 3 SALT : SHARING ATTENTION BETWEEN LINEAR LAYER AND TRANSFORMER

### 3.1 SALT ARCHITECTURE

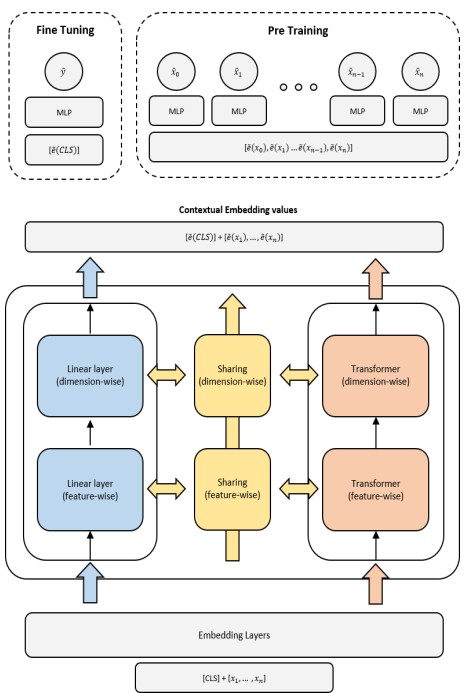

Figure 1: The Architecture of SALT that has two blocks, Transformers and Linear layers. SALT train with [CLS] token for fine-tuning and with contextual embedding values for Semi-supervised learning and Pre-training.

In this section, we introduce our model, SALT (Sharing Attention between Linear layer and Transformer) shown in Figure 1. SALT uses the input embedding values obtained from the embedding

layer. The embedding values have the shape of 'features($n$) $\times$ embedding dimensions($d$)'. The main body of SALT has two blocks: Transformers block inspired by (Vaswani et al., 2017) and Linear layer block inspired by (Liu et al., 2021). Each block has subblocks by feature-wise and dimension-wise. These two subblocks allow communications between different features and different embedding elements and makes the model robust(Tolstikhin et al., 2021). The output values from two blocks become the contextual embedding values. SALT performs fin-tuning and pre-training with contextual embedding values.

### 3.2 SALT FOR LEARNING

The learning process of SALT is as follows. Let $\mathcal{D} = \{x_i, y_i\}_{i=1}^{m}$ be tabular dataset of $m$ samples. The feature($n$) variables $x_{features} \in \mathbb{R}^n$ include categorical $x_{cat}$ and continuous variables $x_{cont}$. SALT adds the special token [cls] to the feature variables and takes them as an input values like BERT (Devlin et al., 2019). So, $x_i = [[\text{cls}], x_{cat}, x_{cont}]$ is the input values consisting of the feature variables and a special token [cls]. The embedding layer $\mathbf{E}(\cdot)$ converts the input values $x_i \in \mathbb{R}^{(n+1)}$ into $d$-dimensional values $\mathbf{E}(x_i) \in \mathbb{R}^{(n+1) \times d}$. Let Transformer be Transformer($\cdot$) and Linear layer block be Linearlayer($\cdot$). SALT consists of an $L$ stack of these two blocks. Each block returns the output value $z$ and the sharing attention matrices, $s_f$ and $s_d$ by feature-wise and dimension-wise as the following equations :

$$z_t^{(1)}, s_{t_f}^{(1)}, s_{t_d}^{(1)} = \text{Transformer}_1(E(x_i)) \tag{5}$$

$$z_l^{(1)}, s_{l_f}^{(1)}, s_{l_d}^{(1)} = \text{Linearlayer}_1(E(x_i)) \tag{6}$$

$z_t^{(1)}$ and $z_l^{(1)}$ are the output values of Transformer block and Linear layer block from the first stack, respectively. $s_{f_t}^{(1)}$ and $s_{d_t}^{(1)}$ are the feature-wise attention matrix and dimension-wise attention matrix from Transformer block. $s_{f_l}^{(1)}$ and $s_{d_l}^{(1)}$ are from Linear layer block. The matrices from blocks are calculated in the head direction by function $S(\cdot)$ and the calculated matrix becomes the sharing attention matrix for the next blocks.

$$S_f(s_{f_t}, s_{f_l}) = \text{concat}(s_{f_t}, s_{f_l})W_f \tag{7}$$

$$S_d(s_{d_t}, s_{d_l}) = \text{concat}(s_{d_t}, s_{d_l})W_d \tag{8}$$

$W_f, W_d \in \mathbb{R}^{h \times 2h}$ are weight for projection in the head direction. The two attention matrices obtained from the above equation are sent to the blocks of the next stack. This procedure is repeated until the last stack as the following equations.

$$z_t^{(i)}, s_{f_t}^{(i)}, s_{d_t}^{(i)} = \text{Transformer}_i(z_t^{(i-1)}, \tilde{s}_f^{(i-1)}, \tilde{s}_d^{(i-1)}) \tag{9}$$

$$z_l^{(i)}, s_{f_l}^{(i)}, s_{d_l}^{(i)} = \text{Linearlayer}_i(z_l^{(i-1)}, \tilde{s}_f^{(i-1)}, \tilde{s}_d^{(i-1)}) \tag{10}$$

$$\tilde{s}_f^{(i)} = S_f(s_{f_t}^{(i)}, s_{f_l}^{(i)}) \qquad \tilde{s}_d^{(i)} = S_d(s_{d_t}^{(i)}, s_{d_l}^{(i)}) \tag{11}$$

Finally, the output values of the last stack, $z_l^L$ and $z_t^L$ are added as the output of SALT. This value also becomes the contextual embedding values $\tilde{E}(x_i) = [\tilde{e}([\text{cls}]), \tilde{e}(x_1), \cdots, \tilde{e}(x_n)]$. The cls token, $\tilde{e}([\text{cls}])$ is used for fine-tuning, and the other values, $\tilde{e}(x_i)$ are used for pre-training using MLM(Masked Language Model)(Devlin et al., 2019).

### 3.3 SHARING ATTENTION

The main idea of SALT is the sharing attention matrices between two blocks. The sharing attention matrices, $\tilde{s}_f$ and $\tilde{s}_d$ are obtained from the function $S(\cdot)$ defined as Equation 7 and 8. The blocks add these matrices to the attention matrix $\frac{QK^T}{\sqrt{d_k}}$ of the self-attention module. This flows similar with (He et al., 2020). The self-attention module returns a new sharing attention matrix $\frac{QK^T}{\sqrt{d_k}} + \tilde{s}$ obtained by adding the attention matrix and the sharing attention matrix $\tilde{s}$. The self attention module works the remaining operations with this new matrix.

$$\text{Attention}(Q, K, V, \tilde{s}) = \text{softmax}(\frac{QK^T}{\sqrt{d_k}} + \tilde{s})V \tag{12}$$

while Transformer has a multi head self-attention module, Linear layer has a single head self-attention module. For calculating the sharing attention matrix, Linear layer repeats the single head attention matrix $\frac{QK^T}{\sqrt{d_k}}$ with the number of heads of the sharing attention matrix.

## 3.4 LINEAR LAYER BLOCK

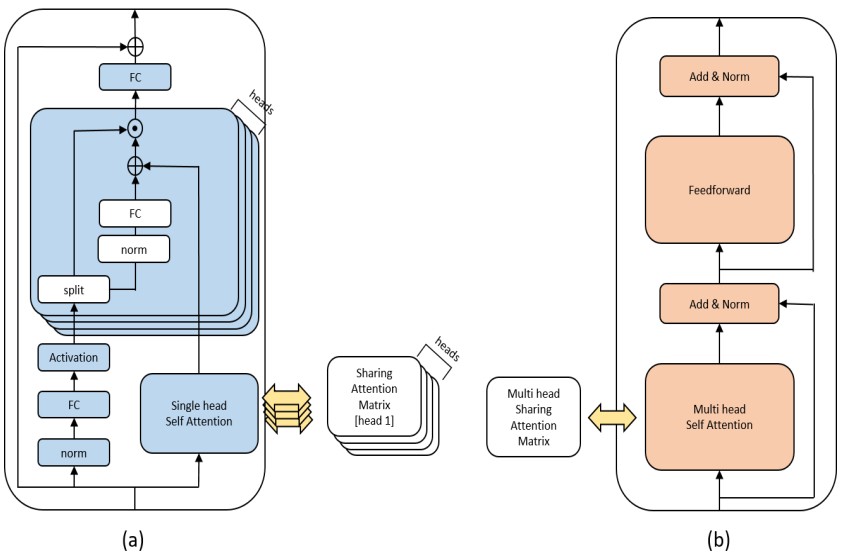

Figure 2: The blocks of SALT. The left (a) is a sub block of Linear layer block. The right (b) is a sub block of Transformer block.

Linear layer block has two sub blocks. The one sub block is forward by feature-wise, the other sub block is by dimension-wise of embedding values. The sub blocks are defined as :

$$attn, s = \text{Attention}(x, \tilde{s}) \tag{13}$$

$$z = \sigma(wx + b) \quad \tilde{z} = g(z, attn) \quad y = w\tilde{z} + b \tag{14}$$

where $\sigma$ is an activation function, $g(\cdot)$ is gating function and $attn$ is the attention value from self-attention module(Liu et al., 2021). The gating function $g(\cdot)$ has two arguments, $z$ and $attn$. To compute with $attn$ which has multi-head form, $z \in \mathbb{R}^{n \times d}$ needs to be converted shape same as $attn \in \mathbb{R}^{h \times n \times d}$. Gating function is defined as :

$$u, v = \text{split}(z) \tag{15}$$

$$v_{norm} = \text{LayerNorm}(v) \tag{16}$$

$$v_{out} = w_v v_{norm} \tag{17}$$

$$\tilde{z} = (v_{out} + attn) \odot u \tag{18}$$

The one of the divisions of $z$, $v$ becomes $v_{out}$ through a normalization and multiplication weights $w_v$. Then, $v_{out}$ is added with the attention value $attn$. Lastly, it calculates element-wise multiplication with $u$, the other of divisions. Linear layer block can capture representations of relationships using the gating function with attention values. Because gating mechanism controls the flow of information and works similar with self-attention. It is expected that the attention value will help capture more representations by adding it to $v$, one of the divided values before element-wise multiplication.

## 3.5 EMBEDDING LAYER OF SALT

SALT introduces the improved embedding layer of continuous variables of tabular data. In our knowledge, there are two methods of embedding for continuous variables. The one is using the embedding matrix $m \in \mathbb{R}^{h \times d}$, projection layers $f(x_{cont_i}) \in \mathbb{R}^{1 \times h}$ and softmax function (Guo

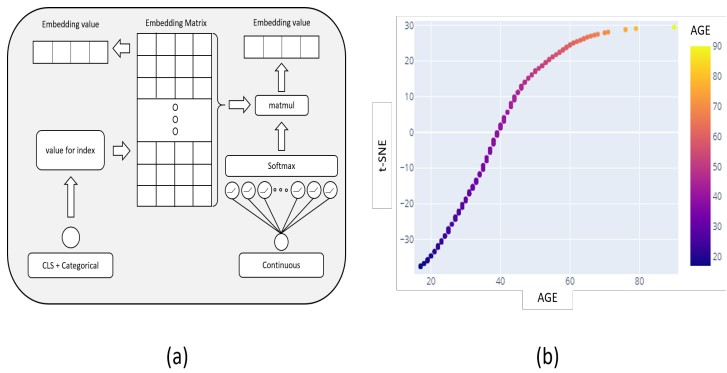

Figure 3: The improved embedding layer introduced by SALT.

et al., 2021b). The other method is the embedding layer that converts $f(x_{cont_i}) \in \mathbb{R}^{1 \times d}$ into $\mathbf{E}_{cont_i}(x_{cont_i}) \in \mathbb{R}^{1 \times d}$ using only the projection layer used in Somepalli et al. (2021). The first method uses different embedding matrices for each variable to obtain an embedding value. SALT improves this method to use the same embedding matrix between categorical variables and continuous variables. As shown in 3 (a), this embedding layer $\mathbf{E}(\cdot)$ works as follows.

$$\mathbf{E}(x_{cont_i}) = \text{softmax}(f_i(x_{cont_i})) \otimes e \tag{19}$$

$$\mathbf{E}(x_{cat_i}) = e[\tilde{x}_{cat_i}] \tag{20}$$

where $e \in R^{hd}$ is embedding matrix, $f_i(\cdot)$ is projection into the $h$-dimensional space $\mathbb{R}^{1 \times h}$ and $\tilde{x}_{cat}$ is the value for the index of embedding matrix $e$. Figure 3 (b) shows the tendency to increase of transformed the embedding value as the original input value increases. The value is the age variables of income dataset which we evaluate in our experiments. We demonstrate that our embedding layer of continuous variables works well and perform better than other methods. See Table 2.

## 4 EXPERIMENTS

### 4.1 BASELINE MODELS

We compare the proposed SALT architecture to the tree-based models such as Decision tree, Random Forests, XGBoost and LightGBM. We also evaluate the deep networks, simple MLP, TabNet and SAINT for comparing. TabNet and SAINT are the deep learning models studied for tabular data learning. SAINT has three variants SAINT, SAINT-i and SAINT-s. Therefore we evaluate the nine models for comparing with our model SALT.

### 4.2 VARIANTS OF SALT

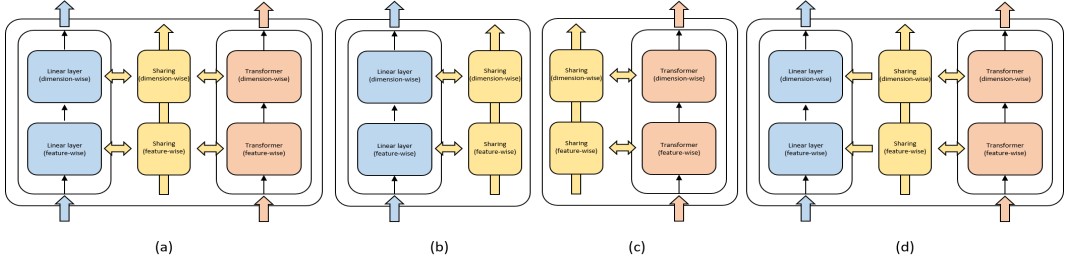

Figure 4: The variants of SALT. (a) SALT (b) SALT-linear (c) SALT-former (d) SALT-oneway

As shown in Figure 4, there are four variants of SALT. The first is two-way sharing between two blocks. The second and third variants are designed only with Transformer and Linear layer blocks,

| share mode | SALT variants | | | |
|---|---|---|---|---|
| | SALT | SALT-oneway | SALT-former | SALT-Linear |
| share | **0.9094** | 0.9075 | 0.9069 | 0.9066 |
| Non share | 0.9089 | 0.9071 | 0.9065 | 0.9055 |

Table 3: The results of the comparison between share mode and Non share mode with SALT variants.

respectively. The last is one way sharing from Transformer block to Linear layer block. We evaluate all of them in the various experiments.

### 4.3 DATASETS AND METRIC

We evaluate the models including SALT on six binary classification datasets which are the various size of samples and features. The size of data samples varies from 1,055 to 45,211 samples. The number of features ranges from 11 to 57. The datasets are divided into 65%, 15% and 25% for training, validation and test sets. The datasets are publicly available in UCI(Dua & Graff, 2017), AutoML(Guyon et al., 2019) and Kaggle. We use the auroc as metric to measure the performance of binary classification. With the six datasets, we evaluate a variety of experiments, including supervised learning, unsupervised learning, and semi-supervised learning as well as the embedding method experiment.

### 4.4 PARAMETERS

We set default values for hyper-parameters as follows. Embedding size is 32, attention dimension is 16, the number of heads is 8 and the depth of stacks is 6. We also set the hyper-parameters of SAINT-s with these values. However, for SAINT and SAINT-i, we set the values following (Somepalli et al., 2021). Because the inter-sample attention of SAINT requires heavy resources for learning, as shown in 1 As optimizer, we use AdamW(Loshchilov & Hutter, 2019) with $\beta_1 = 0.9$, $\beta_2 = 0.999$, decay = 0.01 and learning rate = $1e^{-4}$. We set the 100 epochs, 30 early stopping count and 256 size of batch. For training, we use one GPU of Nvidia GeForce RTX 2080Ti.

| Model | Batch | L | h | Param × 1e6 |
|---|---|---|---|---|
| SAINT | 256 | 1 | 8 | 196.16 |
| SAINT-i | 256 | 1 | 8 | 195.87 |
| SAINT-s | 256 | 6 | 8 | 53.48 |
| SALT | 256 | 6 | 8 | 77.65 |
| SALT-oneway | 256 | 6 | 8 | 77.64 |
| SALT-former | 256 | 6 | 8 | 77.49 |
| SALT-linear | 256 | 6 | 8 | 77.32 |

Table 1: The parameters comparing to SAINT on six datasets

## 5 RESULTS

### 5.1 EMBEDDING LAYERS

We evaluate the three types of embedding layer on six datasets. As mentioned in Section 3.5, the first method is using the multiplication with embedding matrices and values projected by layers. The second is using simple mlp for projection. Our proposed embedding method improves the first method and outperforms the others. See Table 2.

| Methods | Mean AUROC |
|---|---|
| Guo et al. (2021a) | 0.9066 |
| Simple MLP | 0.9079 |
| Ours | **0.9094** |

Table 2: The Mean AUROC for the embedding methods with SALT on six datasets.

### 5.2 SHARING ATTENTION MATRIX

We visualize the attention matrices in Figure 5. It shows the effect of sharing attention matrices. (a) shows the attention matrices, the matrix of the Transformer block on the left, the matrix of the Linear layer block in the middle, and the matrix of the sharing function $S(\cdot)$ on the right. On the other hand, (b), (c) and (d) are models without the sharing attention matrices. They are

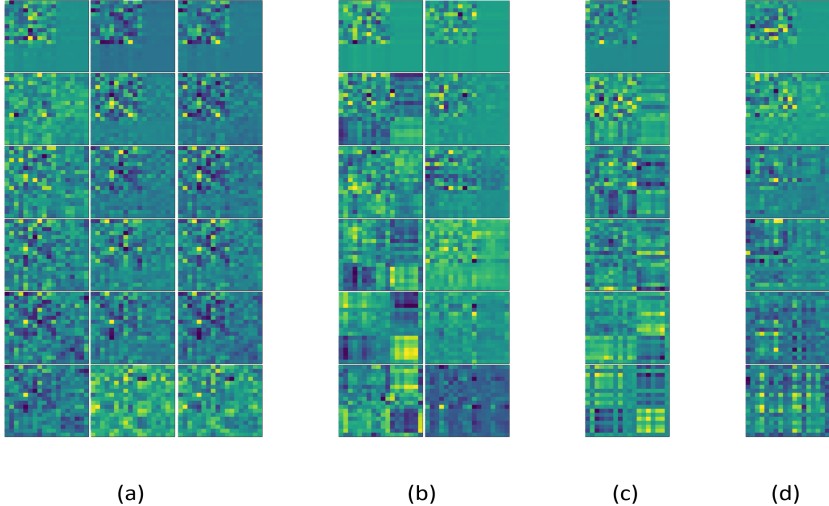

(a)                    (b)                    (c)                    (d)

Figure 5: Visualization of the (feature-wise) attention matrices. For each stack in the model we plot the row. (a) and (b) show the effect of share mode and non-share mode, respectively. (c) is from SALT-foremr and (d) is from SALT-linear.

non-share mode of SALT, SALT-former, and SALT-linear, respectively. They show that each block learns certain parts more intensively. However, (a) shows that the sharing attention matrix helps each block learn more diversely. The matrices are from the experiments of online shopper datasets. The model of (a) has the best performance with an AUROC score of 0.927. (b), (c) and (d) perform 0.925, 0.924, and 0.925 with AUROC scores, respectively.

## 5.3 Semi-supervised Learning & Pre-training

We evaluate SALT, SALT-variants and SAINT under the semi-supervised learning scenario where small-labeled samples are available. We compare five models under four conditions according to the number of labeled samples. The five models are four variants of SALT and SAINT. See Table 4 for results of the AUROC score. In the condition of using all samples, SALT performs the best. Pre-trained SALT and pre-trained SALT-oneway perform best under different conditions with few datasets of 50, 100, and 300 labeled samples. Other variants also show the better performance with pre-training in small data scenarios. We demonstrate that SALT performs well on small data environments with semi-supervised learning.

| | # Labeled | | | |
|---|---|---|---|---|
| Model name | 50 | 100 | 300 | ALL |
| SAINT | 0.7935 | 0.8235 | 0.8549 | 0.9065 |
| SALT | 0.7983 | 0.8251 | 0.8647 | **0.9094** |
| SALT-oneway | 0.7934 | 0.8225 | 0.861 | 0.9075 |
| SALT-former | 0.7909 | 0.8219 | 0.8582 | 0.9069 |
| SALT-linear | 0.7934 | 0.8168 | 0.8587 | 0.9066 |
| SAINT+pt | 0.8025 | 0.827 | 0.8591 | 0.9056 |
| SALT+pt | **0.8068** | 0.8284 | 0.8656 | 0.9084 |
| SALT-oneway+pt | 0.8036 | **0.8288** | **0.866** | 0.9079 |
| SALT-former+pt | 0.8012 | 0.8225 | 0.8588 | 0.9066 |
| SALT-linear+pt | 0.7981 | 0.817 | 0.8589 | 0.9059 |

Table 4: The results under semi-supervised learning, varying by the number of labeled training samples. The pt means pre-training.

## 5.4 Supervised Learning

In Table 5, we report the results on the six binary classification datasets. The models perform with five trials of different seeds. The one of SALT variants outperforms the most models on a variety of datasets. SALT has the best score in four of the six datasets, a majority, compared to other models including GBDTs. It also showed the best performance in the datasets except for one dataset compared to the deep learning models. SALT showe the best performance with an average score in all six datasets.

| Sample size | 4,601 | 10,000 | 7,043 | 1,055 | 32,561 | 45,211 | |
|---|---|---|---|---|---|---|---|
| Feature size | 57 | 11 | 20 | 41 | 14 | 16 | Mean |
| Model \dataset | Spambase | Shrutime | Blastchar | Qsar bio | Income | Bank marketing | |
| Decision Tree | 0.9643 | 0.6698 | 0.8277 | 0.92 | 0.7204 | 0.8051 | 0.8179 |
| Random Forest | 0.9851 | 0.8527 | 0.8196 | 0.9269 | 0.9028 | 0.9254 | 0.9021 |
| XGBoost | 0.987 | 0.8479 | 0.8119 | 0.9216 | 0.9198 | 0.928 | 0.9027 |
| LightGBM | **0.9882** | 0.8611 | 0.8252 | 0.9224 | **0.9236** | 0.9307 | 0.9085 |
| MLP | 0.9758 | 0.4846 | 0.792 | 0.9238 | 0.6226 | 0.7347 | 0.7556 |
| TabNet | 0.9777 | 0.8274 | 0.7923 | 0.8251 | 0.9066 | 0.9196 | 0.8748 |
| SAINT | 0.9835 | 0.8638 | 0.8339 | 0.9198 | 0.9113 | 0.9264 | 0.9065 |
| SAINT-i | 0.9833 | 0.8617 | 0.8327 | 0.9260 | 0.9098 | 0.9259 | 0.9066 |
| SAINT-s | 0.9821 | 0.8584 | 0.8335 | 0.9121 | 0.9124* | 0.9302 | 0.9048 |
| SALT | 0.985* | **0.8675*** | 0.8341 | **0.9272*** | 0.9113 | **0.9311*** | **0.9094*** |
| SALT-oneway | 0.9829 | 0.866 | **0.8347*** | 0.9221 | 0.9109 | 0.9286 | 0.9075 |
| SALT-former | 0.9835 | 0.8634 | 0.8329 | 0.9221 | 0.9104 | 0.9288 | 0.9069 |
| SALT-linear | 0.9832 | 0.8643 | 0.8333 | 0.9199 | 0.9104 | 0.9287 | 0.9066 |

Table 5: The Mean AUROC of evaluating models with six datasets. Bold means the best performance in the entire models. * means the best performance in the deep learning models.

## 6 LIMITATIONS AND CONCLUSIONS

We propose SALT, a novel and hybrid deep learning architecture for tabular learning. SALT introduces the improved embedding method and the sharing attention matrix between two blocks. Furthermore, SALT can be improved for training resource with efficient transformers (Tay et al., 2020). In addition, variants of SALT showed superior performance in four out of six datasets compared to other models including GBDTs. SALT also shows benefits of unsupervised pre-training. With good performance and various benefits of attention matrix, SALT shows the attention mechanism works on tabular datasets too. However, our model still shows lower performance than GBDTs in some data. Therefore, continuous research on tabular data modeling is needed. The code is available at https://github.com/Juseong03/SALT.

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
