# OpenReview forum: "SALT : Sharing Attention between Linear layer and Transformer for tabular dataset"
_ICLR.cc/2022/Conference — ICLR 2022 Submitted_

### Official Review · Reviewer_A8EH · 2021-11-01

**Correctness:** 3
**Technical Novelty And Significance:** 2
**Empirical Novelty And Significance:** 2
**Recommendation:** 5
**Confidence:** 4

**Main Review:**

**Strengths**

- I think it is interesting to develop deep models for tabular data. Learning on tabular datasets is still an under-explored research area.
- The idea of sharing attention matrices looks interesting.
- The authors show that combining a Transformer-like architecture with a gMLP-like one improves the performances. (I assume the SALT results are significantly better than SALT-former and SALT-linear, see my comment later).
- The authors analyzed different variants of their model.

**Weaknesses**

- Section 3.5: “SALT improves this method to use the same embedding matrix between categorical variables and continuous variables.” I am not sure what is the intuition behind this idea. The authors should motivate this claim.
- The model is presented to work on any tabular datasets. However from experience, a lot of tabular datasets contain at least one date/time column. I wonder if SALT can work on date/time columns.
- The model is evaluated only for binary classification tasks. I think adding experiments for other tasks (e.g. multi-class classification, multi-label classification, regression) can increase the quality of the paper.
- The authors should explain how they do the semi-supervised learning in section 5.3.
- For most of the tables, I recommend showing the variance for different random seeds or some other confidence interval measures. For a lot of results, the difference between two models looks small and it is difficult to know if the difference is significant.
- It is difficult to understand Table 3 without additional information. I recommend adding information about the dataset or metric used in the caption. This Table is not mentioned in the text.
- The authors visualize the attention matrices for different variants of their model. However, these visualizations are not commented on so it is difficult to understand the message from this analysis.
- The authors should comment on the scalability of their model. The SAINT model was evaluated on a larger dataset. I wonder why SALT was not evaluated on this dataset. I think the authors should also compare the training time/inference time of their model with existing models.
- The authors should add information about the Transformer architecture used. Is it the original Transformer architecture also called PostNorm? Or is it more recent architecture like PreNorm or ReZero?
- The last sentence of the conclusion breaks the anonymity of the paper: “The code is available at https://github.com/Juseong03/SALT”


**Summary Of The Paper:**

This paper introduces a Transformer-based architecture for tabular datasets. This architecture combines a Transformer with a gating MLP (gMLP) by sharing the attention matrices. The authors also proposed a new approach to encode continuous variables. The proposed model is evaluated on six binary classification tabular datasets. The proposed model is evaluated in a supervised learning context, but also in a semi-supervised context.

**Summary Of The Review:**

Overall, I feel it is quite difficult to understand the potential impact of this paper. The idea of sharing attention matrices looks interesting but there are some questions about the scalability of the proposed approach and if it can work on any tabular datasets.

---

### Official Review · Reviewer_6yF7 · 2021-11-03

**Correctness:** 3
**Technical Novelty And Significance:** 2
**Empirical Novelty And Significance:** 2
**Recommendation:** 5
**Confidence:** 4

**Main Review:**

Pros:

- This manuscript attempts to propose a deep learning model for tabular data, a problem with practical application value.
- An interesting hybrid network architecture is presented, the main idea of which is good and reasonable.

Cons:

- The novelty is limit. The key block SHARING ATTENTION shares similar idea with existing works, RealFormer.
- The writing of the manuscript can be polished. Typos in Equation 7 and 8 are confusing.

**Summary Of The Paper:**

This paper proposed a hybrid deep network architecture for tabular data, dubbed SALT (Sharing Attention between Linear layer and Transformer).
There are two blocks in SALT: Transformers and linear layer blocks. And sharing attention matrices are introduced to promote cooperation between these two blocks.

**Summary Of The Review:**

The proposed SALT, in my view, should be technically correct.
But I think the novelty is limit.
Consequently, although both the problem and the method is okay, this manuscript is a borderline work.
I recommend rejecting it.

---

### Official Review · Reviewer_mutc · 2021-11-08

**Correctness:** 3
**Technical Novelty And Significance:** 2
**Empirical Novelty And Significance:** 2
**Recommendation:** 3
**Confidence:** 3

**Main Review:**

The main strength of the paper is marginal improvement over the SOTA deep learning model SAINT for supervised setting scenario.

The main weakness is in lack of justification for the proposed choice of SALT model, and inadeqate explanation of the SALT architecture. It is not clear what is dmension-wise attention. There is no motivation/justification for the Transformers and Linear layers with attention sharing. Further, why is that there is only one layer of feature-wise attention and linear layer, and then L layers of dimension-wise attention and linear layer. Neiher has been given any good justification for sharing attention between Transformer and Linear layer. Further, for the most important supervised learning scenario, the SALT model barely beats the tree-based LightGBM model.

**Summary Of The Paper:**

This paper proposes a hybrid deep network, named SALT (Sharing Attention between Linear layer and Transformer). The SALT consists of two blocks: Transformer and linear layer blocks that take advantage of shared attention matrices. They compare SALT with tree-based ensemble models and previous deep learning models on multiple benchmark datasets. It furher shows robustness of the proposed SALT with semi-supervised learning and pre-training with small dataset scenarios.

The main body of SALT has two blocks: Transformers block inspired by (Vaswani et al., 2017) and Linear layer block inspired by (Liu et al., 2021). Each block has subblocks by feature-wise and dimension- wise. These two subblocks allow communications between different features and different embed- ding elements and makes the model robust (Tolstikhin et al., 2021). The output values from two blocks become the contextual embedding values. SALT performs fin-tuning and pre-training with contextual embedding values.

For supervised learning setting, the mean AUROC score of the proposed SALT improves upon the treebased LightGBM by mere 0.09%, and it improves upon the best deep learning based model SAINT by 0.29%

**Summary Of The Review:**

Though the numerical exeriments show that the proposed model SALT marginally improves over the sota deep learning model SAINT, it barely beats the tree-based LightGBM model. The architectural choices of SALT have no motivation/justification.

---

### Official Review · Reviewer_eW8f · 2021-11-09

**Correctness:** 2
**Technical Novelty And Significance:** 2
**Empirical Novelty And Significance:** 2
**Recommendation:** 3
**Confidence:** 4

**Main Review:**

I have serious concerns about the paper, as listed below:

- The paper writing is very poor. There are numerous language and grammar errors. At many places, the expressions are not clear. The quality of the figures is low.

- Overall, the novelty is low. Transformer architecture is modified in a straightforward way. Even the semi-supervised learning approach is the simple adaption of MLM-like unsupervised pre-training.

- Benchmarking is employed and presented very poorly. For comparison models like TabNet, XGBoost etc., it is not clear how the hyperparameter tuning is done and what parameters are included in the search space. The authors should clearly describe which parameters were tuned and what validation reward is used. Otherwise, the outperformance conclusions are not convincing at all.

- How do you do hyperparameter tuning in semi-supervised regime? It is unclear how semi-supervised validation data is used.

- The results are presented in AUROC but what is the training objective? If the training objective is not AUROC, how do you ensure the metric mismatch is not dominating?

- What is the significance of Fig. 5? How can you convince the readers that SALT has learned attention patterns that are meaningful?

- The difference between supervised learning results is very low across different models. It is unclear whether the results are statistically significant.

- No ablation studies are presented for the major constituents of the claims, such as the benefit of sharing attention layers.

**Summary Of The Paper:**

The paper proposes a new tabular deep learning architecture based on sharing attention matrices enable transformers and linear layers. Comparisons with other tabular learning models on various benchmarks are demonstrated.

**Summary Of The Review:**

I suggest substantially revising the paper and submitting to another venue.

---

### Decision · Program_Chairs · 2022-01-20

**Decision:**

Reject

**Comment:**

This paper proposes a method to use Transformers with tabular data by sharing attention. Reviewers raise significant concerns about the motivation, writing and experimental results. Author's did not submit a response. Hence I recommend rejection.